# Gait Characteristics and Cognitive Function in Middle-Aged Adults with and without Type 2 Diabetes Mellitus: Data from ENBIND

**DOI:** 10.3390/s22155710

**Published:** 2022-07-30

**Authors:** Pieter M. R. Herings, Adam H. Dyer, Sean P. Kennelly, Sean Reid, Isabelle Killane, Louise McKenna, Nollaig M. Bourke, Conor P. Woods, Desmond O’Neill, James Gibney, Richard B. Reilly

**Affiliations:** 1School of Engineering, Trinity College Dublin, D08 XW7X Dublin, Ireland; pieter.herings@gmail.com (P.M.R.H.); seanoreid@gmail.com (S.R.); reillyri@tcd.ie (R.B.R.); 2Trinity Centre for Biomedical Engineering, Trinity College Dublin, D02 R590 Dublin, Ireland; 3Department of Medical Gerontology, School of Medicine, Trinity College Dublin, D02 R590 Dublin, Ireland; sean.kennelly@tuh.ie (S.P.K.); nbourke@tcd.ie (N.M.B.); desmond.oneill@tuh.ie (D.O.); 4Department of Age-Related Healthcare, Tallaght University Hospital, D24 NR0A Dublin, Ireland; mckennl1@tcd.ie; 5Department of Engineering, Technological University Dublin, D07 EWV4 Dublin, Ireland; isabelle.killane@tudublin.ie; 6Robert Graves Institute of Endocrinology, Tallaght University Hospital, D24 NR0A Dublin, Ireland; conor.woods@hse.ie (C.P.W.); james.gibney@tuh.ie (J.G.)

**Keywords:** dementia, mild cognitive impairment, gait, prediction models, neural network

## Abstract

Type 2 Diabetes Mellitus (T2DM) in midlife is associated with a greater risk of dementia in later life. Both gait speed and spatiotemporal gait characteristics have been associated with later cognitive decline in community-dwelling older adults. Thus, the assessment of gait characteristics in uncomplicated midlife T2DM may be important in selecting-out those with T2DM at greatest risk of later cognitive decline. We assessed the relationship between Inertial Motion Unit (IMUs)-derived gait characteristics and cognitive function assessed via Montreal Cognitive Assessment (MoCA)/detailed neuropsychological assessment battery (CANTAB) in middle-aged adults with and without uncomplicated T2DM using both multivariate linear regression and a neural network approach. Gait was assessed under (i) normal walking, (ii) fast (maximal) walking and (iii) cognitive dual-task walking (reciting alternate letters of the alphabet) conditions. Overall, 138 individuals were recruited (*n* = 94 with T2DM; 53% female, 52.8 ± 8.3 years; *n* = 44 healthy controls, 43% female, 51.9 ± 8.1 years). Midlife T2DM was associated with significantly slower gait velocity on both slow and fast walks (both *p* < 0.01) in addition to a longer stride time and greater gait complexity during normal walk (both *p* < 0.05). Findings persisted following covariate adjustment. In analyzing cognitive performance, the strongest association was observed between gait velocity and global cognitive function (MoCA). Significant associations were also observed between immediate/delayed memory performance and gait velocity. Analysis using a neural network approach did not outperform multivariate linear regression in predicting cognitive function (MoCA) from gait velocity. Our study demonstrates the impact of uncomplicated T2DM on gait speed and gait characteristics in midlife, in addition to the striking relationship between gait characteristics and global cognitive function/memory performance in midlife. Further studies are needed to evaluate the longitudinal relationship between midlife gait characteristics and later cognitive decline, which may aid in selecting-out those with T2DM at greatest-risk for preventative interventions.

## 1. Introduction

Dementia is expected to affect 65.7 million individuals globally by 2030 [1]. Whilst new medications for the management of dementia have been limited [2], findings from multi-domain non-pharmacological trials are encouraging and emphasize the importance of early risk-factor modification in at-risk individuals [3]. Many of the risk factors for the later development of dementia exert their risk in midlife, such as hypertension, obesity, hypercholesterolaemia and Type 2 Diabetes Mellitus (T2DM) [4,5]. T2DM appears to act as a risk factor for the later development of dementia during the fourth to sixth decades of life, with the association somewhat attenuated thereafter [1]. Both the duration of T2DM and the presence of micro- and macrovascular complications of T2DM appear to be associated with later dementia risk [6]. Thus, early identification of individuals free from T2DM in midlife who may be at greater risk of later cognitive decline may aid in selecting-out those who may benefit most from multidomain preventative interventions, which have shown promise in delaying the onset of dementia in other at-risk groups [7].

There is now a wealth of evidence to suggest that gait characteristics (and in particular gait speed) is one of the most promising non-invasive biomarkers of later cognitive decline in middle-aged and older adults [8,9,10,11,12]. For instance, in the English Longitudinal Study on Ageing (ELSA) both walking speed at baseline and a decline in walking speed were both associated with a greater risk of dementia [13]. Similarly, in data from the Gait and Brain Study, a decline in gait speed was associated with a seven-fold greater risk of subsequent dementia [14]. In addition to gait speed in isolation, other spatiotemporal gait characteristics such as step variability and asymmetry have been associated with cognitive function, and in particular attention and executive function domains [15]. Further, the addition of a dual-task (such as reciting alternate letters of the alphabet whilst walking) has been shown to introduce significant executive function and attention demands and may be a particularly useful marker of later cognitive decline [16], particularly when measured at multiple time-points in the same individual [17,18].

Traditionally, gait speed has been measured using a stopwatch and runway. The variables obtained by this method are easy to obtain, but prone for human errors. Electronic walkways use pressure sensors in a mat to measure both temporal and spatial gait variables. They are known for their excellent validity and reliability [19,20], but are relatively expensive and may be impractical in certain clinical contexts. Gait can also be measured with accelerometer data obtained by Inertial Motion Units (IMUs) on the lower limbs, followed by signal analysis, to extract a range of gait characteristics. IMUs are known to be low cost, convenient and portable, making them very user-friendly. In particular, IMU devices such as Shimmer^®^ containing a tri-axial gyroscope to assess temporal gait characteristics offer a flexible, wireless platform for the measurement of gait characteristics and have been previously validated in both younger and older adults [21,22,23]. Such devices offer a pragmatic approach to assessing gait in busy clinical contexts, where use of electronic walkways may be impractical.

Importantly, whilst many studies assessing the relationship between gait speed and cognitive function have been carried out in older adults, relatively few studies have examined the relationship between midlife gait characteristics and cognitive function [24]. In those with T2DM, studies examining the relationship between gait and cognitive function have been largely limited to older adults and those with established T2DM complications such as peripheral neuropathy [25]. Similarly, studies assessing the impact of T2DM on gait performance are again largely limited to older adults and those with established complications/neuropathy [26,27,28,29]. Thus, the relationship between early subtle decrements in gait performance and subtle cognitive decrements in midlife T2DM, at the precise time when T2DM is acting as a risk factor for later cognitive dysfunction, is not currently clear. Understanding the relationship of subtle decrements in gait and cognitive function in midlife may aid in selecting-out those most at risk of cognitive decline for potential preventative interventions.

In the current study, we aimed to assess the impact of uncomplicated T2DM on IMU-derived gait characteristics across three tasks in middle-aged adults free from objective cognitive impairment, in addition to the potential relationships between IMU-derived gait characteristics and cognitive function. Finally, we compare the predictive performance of both traditional multivariable linear regression models in comparison with a neural network approach to assess the relationship between gait characteristics and cognitive function in midlife T2DM.

## 2. Materials and Methods

### 2.1. Participants

ENBIND (Exploring Novel Biomarkers of Brain Health in Midlife Type 2 Diabetes) is a longitudinal study of cognition in midlife Type 2 Diabetes Mellitus (T2DM) [30,31]. Middle-aged adults with T2DM and healthy controls were recruited from a T2DM clinic located in a tertiary referral hospital (Tallaght University Hospital, Dublin, Ireland) and by local advertisement within the same hospital in a 2:1 ratio.

Inclusion criteria for those with T2DM included a confirmed diagnosis of T2DM (physician-diagnosed, recruited from a tertiary T2DM clinic in TUH) free from any microvascular or macrovascular complications and aged 35–65 years of age. Individuals with non-type 2 diabetes, evidence of any diabetes-related complications (neuropathy, nephropathy, cerebrovascular, cardiovascular or peripheral vascular disease) were excluded. For both groups, individuals with current depression (a score of 8 or greater on the Centre for Epidemiological Scale 8) or significant cardiorespiratory, musculoskeletal, psychiatric or neurological comorbidity were excluded from participation. Healthy controls were screened for diabetes (defined as a HbA1c above the cut-off value for a diagnosis of diabetes). All participants with a score of <23 on the Montreal Cognitive Assessment were excluded from further participation in order to assess only individuals free from pre-existing cognitive impairment. Participants were additionally screened using the Diabetes Neuropathy Symptom Score (DNSS) and any individual with T2DM and a score of two or greater indicating those at risk of, or with early signs of peripheral neuropathy, excluded from participation.

### 2.2. Cognitive Assessment

#### 2.2.1. Montreal Cognitive Assessment

General cognitive function was assessed using the Montreal Cognitive Assessment (MoCA) [32], a short assessment of cognitive function used both clinically and in population studies for the detection of cognitive impairment. MoCA subdomains include: visuospatial/executive function, naming, attention, language, abstraction, delayed memory and orientation. A one-point addition is made for 12 or fewer years of formal education. As above, any participant with a score of less than 23 was excluded from participation.

#### 2.2.2. Detailed Neuropsychological Assessment

Detailed neuropsychological assessment was carried out following administration of the MoCA. Briefly, individuals were administered a comprehensive custom-battery of 6 tests lasting just over 60 min in duration. Tasks consisted of: (i) Reaction Time Task, (ii) Paired-Associates Learning (a test of short-term memory), (iii) Spatial Working Memory, (iv) Pattern Recognition Memory (a test of delayed memory), (v) One-Touch Stockings of Cambridge (a task with significant executive function demands) and Rapid Visual Processing (a task of attention). The six tasks were designed to assess memory (ii–iv), executive function (v) and attention as these domains are amongst the first to be affected in T2DM-related cognitive impairment. Scores on each tasks were Z-transformed to aid interpretability and analysis.

### 2.3. Gait Assessment

#### 2.3.1. Experimental Setup

Two Shimmer^®^ 3 inertial moment sensors (IMU) [33] were used to collect gait data. Shimmer IMUs are lightweight, low-power sensor platform that incorporate a tri-axial accelerometer on the base board with a gyroscope on the daughter board enabling extraction of temporal parameters of gait in clinic/ambulatory settings [23]. Shimmer devices have been widely validated in the detection/assessment of heel-strike and toe-off and in the extraction of stride, stance and swing time in ambulatory gait assessments [23]. The sensors were attached anteriorly to the mid-shin, with Velcro straps. Measures of accelerations in both legs were recorded at either 20 kHz or 8 kHz and later down-sampled to 200 Hz [34]. The IMU recordings were turned on and off manually and docked to the Shimmer base after use. The data was stored on an SD card.

#### 2.3.2. Experimental Protocol

Gait was assessed during three different gait tasks. For each task, participants were asked to walk 10 m, turn around, and walk another 10 m in a hospital corridor (no windows, and lights turned on for all participants). Three different tasks were executed while doing this 20-m walk; (i) “Normal” walking speed, where participants were instructed to walk as they normally do, (ii) “Maximal” gait speed where participants were instructed to walk as fast as they could, without running, (iii) Cognitive dual-task condition, where participants were instructed to walk at their normal pace as under the “Normal” walking speed condition and additionally recite alternating letters of the alphabet (A-C-E etc.) whilst walking.

### 2.4. Obtaining Gait Parameters from Shimmer^®^ IMUs

#### 2.4.1. Standard Gait Parameters

The accelerometer data, obtained from the Shimmer^®^ IMU, in the sagittal plane from each walking task was used to determine standard temporal gait parameters (code in Section A.1, Figure 1a). The timestamps from the major gait events [heel strike (HS), toe-off (TO) and max swing velocity (MSV)] were determined from the raw accelerometer data (Figure 1b). First, the accelerometer data from the walking tasks was manually selected. The manual extraction step included removal of accelerometer data that was recorded before and after the walking task (i.e., walking from the examination room to the starting point or waiting after finishing the walking task). The signal was then down-sampled to 200 Hz and filtered as previously described [21,34,35], using a median filter with window size 5 and a 4th order Butterworth lowpass filter. The result was a smooth signal (Figure 1c), with timestamps of peaks and troughs identified and labeled as max swing velocity, toe-off or heel strike (Figure 1d).

The average walking velocity over the gait task was calculated by dividing the walking distance (20 m) by the total walking time. Before averaging the swing, stance and stride time for each foot, the turning artefact (Figure 2) was removed. Swing, stance and stride times deviating more than two standard deviations from the mean were deleted to ensure removal of the turning artefact (over a complete total of 755 walking tasks, the mean removed strides was 1.01 ± 0.50).

Swing, stance and stride times (Figure 2) from each walk were then calculated from the heel strike (HS), toe-off (TO) and max swing velocity (MSV) using Formulas (1)–(3) [34];
(1)Swing time=HS (n)−TO (n)
(2)Stance time=TO (n+1)−HS (n)
(3)Stride time=MSV (n+1)−MSV (n)

#### 2.4.2. Coefficient of Variability

The coefficient of variability (CoV) (Section A.2) quantifies the variability in step duration Formula 4 was used to calculate the CoV [34];
(4)$$CoV=\frac{Standard\;deviation{\;}_{Stride\;times}}{Mean{\;}_{Stride\;times}}\;\text{×}\;100$$

#### 2.4.3. Complexity Index

The complexity index (Section A.3) was obtained using the raw accelerometer data (200 Hz). Pincus et al. introduced Sample Entropy ($S_{E}$), a set of measures of system complexity closely related to entropy [36]. Multiscale entropy (MSE) builds on the sample entropy technique by integrating a coarse graining (Figure 3a) which provides insight into the point-to-point fluctuations over a range of time scales [37]. The coarse graining procedure was defined as:(5)$$y_{j}^{\tau }=\frac{1}{\tau }\;\overset{j\tau }{\underset{i=\left(j-1\right)\tau +1}{\sum }}x_{i},\;\;1\geq y_{j}\geq \frac{N}{\tau }$$
where *τ* is the timescale of interest, $y_{j}$ is a data point in the newly constructed time series, $x_{i}$ is a data point in the original time series and *N* is the length of the original time series.

Secondly, the concept of $S_{E}$ [37].was employed to calculate an entropy value for each timescale;
(6)$$S_{E}\left(m,\;r,\;N\right)=-lnln\;\frac{\phi^{m+1}\left(r\right)}{\phi^{m}\left(r\right)}\;$$
where *m* is the sample length, *r* is the tolerance, *N* is the length of the time series and *φ* is the probability that point’s *m* distance apart would be within the distance *r*. The sample entropy value for each time scale was calculated and plotted, as visualized in Figure 3b. The area under this MSE curve is the complexity index. Regular and predictable time series generate a low value for the complexity index, where unpredictable time series generate a high complexity index [37].

Three input variables must be considered: the sample length (*m*), tolerance (*r*) and the time series length (*N*). The input variables must be consistent when comparing complexity of time series between participants [38]. From studies reported in the literature [39], the sample length should be fixed at 2 and the tolerance at 0.2 when calculating the complexity of gait data in the sagittal plane. These values were used for all complexity calculations. The MSE has shown to be largely independent of *N* when the analyzed signals consist of more than 750 datapoints [40], thus for all the accelerometer signals. In this gait study *N* = 40 was used for all complexity calculations. The coarse graining procedure (with *N* = 40) from a single walking task is visualized in Figure 3a. The sample entropy was calculated from each time scale (MSE plot, Figure 3b). The area under the MSE plot is the complexity index.

### 2.5. Predicting Cognitive Performance from Gait Data

To establish a simple method for predicting cognitive performance from gait variables, prediction models were trained on the entire dataset (both healthy and uncomplicated T2DM). As stated in the hypothesis, two types of regression models were compared by their predictive accuracy to predict cognitive score with gait variables. The total mean absolute error of the prediction was used as a measure of accuracy. Multivariable linear regression adjusting for age, sex, BMI and years of education was compared to a neural network, as described below. Both models were trained with gait variables that correlate with cognitive performance, determined in the next section.

#### 2.5.1. Variable Selection for Multivariate Linear Regression and Neural Network Analysis

Pearson correlation analysis in the overall cohort was performed to test associations between gait variables with cognitive scores (MoCA and CANTAB). Pearson correlations were calculated in Python 3.8 (scipy.stats.pearsonr (gait variable, cognitive score)). Variables with correlation coefficients that meet the certeria −0.3 ≥ R^2^ ≥ 0.3 were considered a moderate to strong correlation [39] and were used to train the prediction models. A heat map was used to visualize high correlation coefficients.

#### 2.5.2. Multivariable Linear Regression

The multivariable linear regression (Least Squares) was performed in GraphPad Prism 9.0.1. The participants were split (Python 3.8, train_test_split() from sklearn) in a training and validation set (80% training, 20% validation), following the Pareto Principle [41]. The training set was used to determine the parameter estimates and the validation set was used to test prediction ability. The model generated confidence intervals (95%) for the parameter estimates and was displayed as follows;
(7)$$Cognitive\;score=\beta_{1}\times V_{1}+\beta_{2}\times V_{2}\;\ldots +\beta_{n-1}\times V_{n-1}+\beta_{n}\;$$
where *Cognitive score* is the MoCA or CANTAB score, β is a parameter estimate and V is a correlating gait variable (−0.3 ≥ R^2^ ≥ 0.3). The measure of evaluation is the total mean absolute error (MAE), which outperforms the root mean squared error (RMSE) to describe model performance [42]. A 20-fold validation was used to determine the MAE with confidence intervals (95%).

#### 2.5.3. Neural Network

A neural network was developed in TensorFlow [43], a user-friendly open source machine learning platform (Python 3.8, TensorFlow 2.4.1, Keras 2.4.0). The code can be found in Section A.3. Preprocessing included removal of subjects with missing data and standardization of all data. Standardization (z-value) of input variables improves accuracy of neural network predictions [44]. The z-scores of variables were calculated as follows;
(8)$$Z=\frac{x-µ}{\sigma }$$
where *Z* is the z-score, x is the observed value, µ is the mean of the sample and σ is the standard deviation of the sample.

A previously reported neural network architecture to predict energy expenditure from IMU data was adopted in this study to predict cognitive scores [45,46]. The architecture of this multi scale neural network depends on the amount of input variables in the input layer. The input layer consists of *n* correlating gait variables (see Section 2.3.1). Then, fully connected hidden layers with a descending amount of neurons/layer until the output layer is reached with 1 neuron (see Figure 4). Sigmoid activation functions were used in all nodes [46]. Lastly, an optimizer was determined to change the attributes of the neural network such as weights and learning rate to reduce the losses per epoch. The optimizer Adam was used because of its optimal convergence properties and computational efficiency [47].

The model was trained with the dataset for 10,000 epochs, which indicates the number of passes of the entire training dataset. Early stopping of model training was used to reduce overfitting issues [48]. The dataset was split into a training and validation set (80% training, 20% validation), following the Pareto Principle [41]. A 20-fold validation was used to determine the mean absolute error (MAE), as evaluation metric, with confidence intervals (95%).

### 2.6. Statistical Analysis

Statistical analysis was performed in GraphPad Prism 9.0.1. and STATA 17.0. Statistical significance was considered at a threshold of *p* < 0.05. In addition to methods described above, standard descriptive statistics were used for obtaining means, standard deviations and percentages. Shapiro–Wilk test was used to test normality of each dataset. Comparisons between groups were performed with an unpaired student t-test (if parametric) or a Mann–Whitney U test (if non-parametric). Models assessing the impact of T2DM on gait characteristics in the first instance adjusted for age, sex, Body Mass Index (BMI) and education level. In assessing the link between gait characteristics and cognitive function, the confidence intervals after bootstrapping, or random sampling, with 20-fold validation was used to compare the total mean absolute error of the prediction models between standard multivariate linear regression and neural network approaches.

### 2.7. Ethical Approval

Ethical Approval was obtained from the Tallaght-St James’s Joint Research Ethics Committee [Reference: 2018/09/02/2018-10 List 34].

## 3. Results

### 3.1. Patient Characteristics

In total, 94 participants with uncomplicated midlife Type 2 Diabetes Mellitus and 44 healthy controls participated in the study. The baseline characteristics of study participants are given by group in Table 1. Overall, individuals with midlife T2DM had significantly greater BMI (*p* < 0.05) (Table 1). Two participants, both with Type 2 Diabetes were excluded from further participation due to having a MoCA score < 23.

### 3.2. Cognitive Assessment

#### 3.2.1. Montreal Cognitive Assessment (MoCA)

Overall, 137 participants had the MoCA cognitive assessment administered. One healthy control terminated testing early and did not complete the MoCA. Midlife Type 2 diabetes mellitus was associated with a significantly lower total MoCA (*p* < 0.001) score (Table 2). In detailed analysis of MoCA domains, midlife T2DM was associated with poorer performance on visuospatial/executive and delayed recall subtasks of the MoCA.

#### 3.2.2. Neuropsychological Assessment (CANTAB Battery)

Nearly all (136/138; 98.6%) of study participants underwent full CANTAB neuropsychological assessment. Two participants with T2DM did not complete the CANTAB assessment due to early termination of testing. On analyzing cognitive performance between the two groups, only the first attempt of the memory score (PALFAMS 28) significantly differed between the T2DM and healthy controls (Table 3).

### 3.3. Gait Assessment

All 138 participants (44 healthy control and 94 Midlife T2DM) completed the gait assessment. Results of gait assessment providing the 6 different gait parameters are given in Table 4. Overall, midlife T2DM was associated with significantly slower gait speed during all three walking tasks, longer stance and stride time during normal walk, longer stance time and higher complexity index during dual-task walk. On adjustment for age, sex, BMI and education, midlife T2DM remained significantly associated with slower gait speed during normal and fast walking (both *p* < 0.001) in addition to a greater stride time and complexity (both *p* < 0.005) during normal walking (See Table 4; Figure 5).

### 3.4. Predicting Cognitive Performance from Gait Characteristics

In order to analyze the relationship between gait and cognitive function in middle-aged adults with and without T2DM, models were developed to predict cognitive scores with discriminative gait variables as described above. The entire dataset (both participants with uncomplicated T2DM and matched healthy controls) were used to develop prediction models.

#### 3.4.1. Variable Selection

To determine which gait variables have the best discriminative characteristics to predict cognitive function (both MoCA and CANTAB scores), a Pearson correlation coefficient analysis was performed. The correlations of gait variables and MoCA scores are shown in Appendix A, for the left and right foot, respectively. For CANTAB scores, correlations are detailed in Appendix A, for the left and right foot, respectively. A heat map is used to visualize strength of correlation coefficients (Figure 6).

#### 3.4.2. Correlation between Gait Characteristics and Cognitive Function

The total MoCA score correlated with normal walking velocity (0.47/0.33 for left/right foot, respectively), fast walking velocity (0.42/0.43 for left/right foot, respectively) and dual-task walk velocity (0.43/0.43 for left and right foot, respectively). For MoCA sub-scores, correlations were found between visuospatial and executive, language and delayed recall and walking velocity across all three tasks.

Given these results, gait velocity was employed to train the subsequent neural network model and compare performance in assessing the relationship between gait performance and cognitive function (MoCA) in comparison to a multivariate linear regression model.

On analyzing the relationships between domain-specific performance and gait characteristics, we observed significant associations between both paired associate learning performance/pattern recognition memory performance and velocity during the normal walk (both *p* < 0.05). Additionally, reaction time significantly correlated with stance and stride time during both normal and fast walking (all *p* < 0.05). However, none of the associations met the pre-specified criteria (−0.3 ≥ R² ≥ 0.3) for further analysis. (Appendix A). As such, no prediction models were trained to predict CANTAB scores with gait variables.

#### 3.4.3. Development of Prediction Models to Predict MoCA Score from Gait Characteristics

Participants who completed all three normal, fast and dual-task walking tasks and had a total MoCA score available, were employed for the prediction models (133 participants; 44 healthy controls and 89 T2DM). Both models were trained with 20-fold validation and results were averaged.

##### Multivariable Linear Regression

The total mean absolute error (MAE) was used as a measure of evaluation. The multivariable linear regression model resulted in an MAE of 1.42 [95% CI; 1.32–1.52], as seen in Figure 7c. The prediction ability of one of the trained networks is visualized in Figure 7b. Confidence intervals of the parameter estimates are shown in Table 5. Total MoCA score was predicted using Equation (7). Velocity refers to walking velocity, norm is the normal walk, fast is the fast walk, dual is the dual-task walk, L is the left foot and R is the right foot.
$$Total\;MoCA=5.74\times Velocity_{norm_{L}}-4.39\times Velocity_{norm_{R}}+1.03\times Velocity_{fast_{L}}$$
$$+0.61\times Velocity_{fast_{R}}+1.01\times Velocity_{dual_{L}}+1.21\times Velocity_{dual_{R}}+21.2\;$$


##### Neural Network Regression

The average Mean Absolute Error of the neural network based regression model was calculated to be 1.37 [95% CI; 1.35–1.40] as shown in Figure 7c. The prediction ability of one of the trained networks is visualized in Figure 7a. Overall, the use of a neural network approach did not outperform multivariable linear regression models in predicting cognitive function from gait characteristics in middle-aged adults with or without T2DM.

## 4. Discussion

The current study demonstrates the impact of T2DM on gait characteristics in middle-aged-adults free from objective cognitive impairment and T2DM-related complications. Uncomplicated T2DM in midlife was associated with subtle impairments in both gait and cognitive performance, most demonstrably for overall gait speed (across three discrete tasks, which persisted for normal and fast walking following covariate adjustment) and global cognition (assessed using the MoCA). Using a neural network model did not outperform standard multivariate linear regression in assessing the predictive performance of overall gait velocity to predict the mean absolute error on global cognitive assessment. Whilst our findings demonstrate the striking relationship between gait and cognition in otherwise-healthy middle-aged adults with and without T2DM, longitudinal studies are required to evaluate the predictive performance of midlife gait characteristics in identifying those at risk of later cognitive decline. Importantly, in the current analysis, no participants had any evidence of microvascular or macrovascular complications of diabetes and were recruited in midlife, when T2DM is known to act as a risk factor for later cognitive decline.

The current results in assessing the relationship between uncomplicated midlife Type 2 Diabetes Mellitus (T2DM) and cognitive performance are largely in keeping with previous literature, although our study assessed a younger population free from T2DM associated complications [49,50,51]. Furthermore, we demonstrated the significant impact of midlife T2DM on walking speed during normal, fast, and dual-task gait tasks, which whilst demonstrated in previous studies for older adults and those with T2DM-related complications, has never been assessed in middle-aged adults free from T2DM associated complications.

The most robust correlation between gait characteristics and cognitive function in the current study was overall gait velocity and total MoCA score across all three walks. Of note, dual-task assessment did not increase discriminative power of the gait variables, which is in contrast to results obtained from research involving older individuals (>65 years) [36]. In older individuals (>65 years), years of education [52,53], gait velocity [54,55,56] and the coefficient of variability during dual-task [57] were found to be discriminative for cognitive ability and in particular for attention and executive function domains. It is noteworthy that we did not observe similar findings in a midlife cohort. The outcome of this study suggests that future studies may focus on a midlife participant group to determine discriminative gait variables for cognitive ability and prediction of future cognitive decline. This may reflect the fact that our cohort were middle-aged, one of the youngest evaluated in the literature in dual-task gait, and were also free from any established cognitive impairment or T2DM-related complications. In such a healthy population in comparison to other cohorts studied, decrements in dual-task performance may be too subtle to detect using standard assessments.

Our study has several important limitations. Participants in the current study consisted of individuals with T2DM free from microvascular and macrovascular complications of T2DM. However, the analysis of baseline characteristics revealed that participants with T2DM had a significantly higher BMI than healthy controls. However, multivariate regression models in the current study adjusted for BMI [58,59]. Future studies could also include control participants with a higher BMI, if possible, to assess the independent influence of T2DM on gait independent of BMI

Notably, the predictive ability of both models was found to lack the ability to discriminate participants with subtle cognitive decrements from healthy subjects. Whilst the multivariable linear regression model outperformed other predictive models based on Pearson correlation analysis [60,61,62], the prediction for lower MoCA scores was poor. It is important to note that we excluded participants with a score of 23 or less in order to exclude those who may have established cognitive impairment. In future studies, additional gait characteristics should be investigated for a distinctive ability between healthy control and Mild Cognitive Impairment (MCI) in middle-aged and older adults with T2DM (not examined in the current study). As we were looking for the earliest possible signs of gait and cognitive/neuropsychological decrements in midlife T2DM, we did not examine those with established cognitive impairment by design. Further, whilst our findings demonstrate the significant relationship between gait velocity and overall cognitive performance in midlife T2DM, gait characteristics in isolation may not have the predictive performance to identify those with early subtle cognitive decrements.

In the current study, a pragmatic clinical study, we employed temporal gait characteristics readily derived from IMUs. The selection of gait variables was based on previously validated parameters such as step, stride and swing time using the Shimmer^®^ system [23]. Whilst this validated and clinically useful system afforded practical and readily available gait parameters to be extracted, future studies should consider a broader range of spatiotemporal gait characteristics. For instance, spatial characteristics such as stride- and step length should be incorporated into future analysis. Such parameters have previously been reported as discriminative between older individuals with and without MCI and may be a useful marker to explore in at-risk populations in midlife such as those with uncomplicated T2DM [58]. The addition of an extra IMU sensor on the lower back, together with measurements of the pendulum leg length for each participant, would allow for accurate estimations of step length [63,64]. However, use of novel parameters in the current analysis, such as the complexity index during walking, allows additional information to be obtained in a routine clinical context that may be important in considering future cognitive decline.

Whilst our study was cross-sectional in nature, longitudinal studies are crucial in order to identify longitudinal relationships between spatiotemporal gait characteristics and later cognitive decline in midlife T2DM. Longitudinally profiling gait and cognitive trajectories from midlife may identify essential non-invasive biomarkers that predict transition into cognitive impairment and dementia in those with midlife T2DM

## 5. Conclusions

Overall, we demonstrate the association between uncomplicated T2DM and subtle decrements in cognitive and gait performance. Based on the dataset used in this study, a neural network model did not outperform a standard multivariable regression model in order to predict cognitive function assessed by the Montreal Cognitive Assessment (MoCA). Uncomplicated midlife T2DM was associated with poorer cognitive performance. Furthermore, midlife T2DM had a negative effect on walking speed during normal, fast and dual-task walks, and is associated with a higher gait complexity index during normal walking.

The correlation analysis between gait and cognition revealed that gait velocity during normal walk, fast walk and dual-task walk was correlated with total MoCA score in midlife participants. This outcome suggests that the focus in a midlife participant group to determine discriminative variables for cognitive ability should be on gait velocity. The dual-task assessment did not increase discriminative power, which is in contrast to studies with older individuals, although future studies are required. Lastly, a multivariable linear regression model performed with similar accuracy to a neural network when predicting total MoCA score with gait variables. The added intricacy of the neural network regression model in this cohort did not result in predictions with a lower mean absolute error

## Figures and Tables

**Figure 1 sensors-22-05710-f001:**
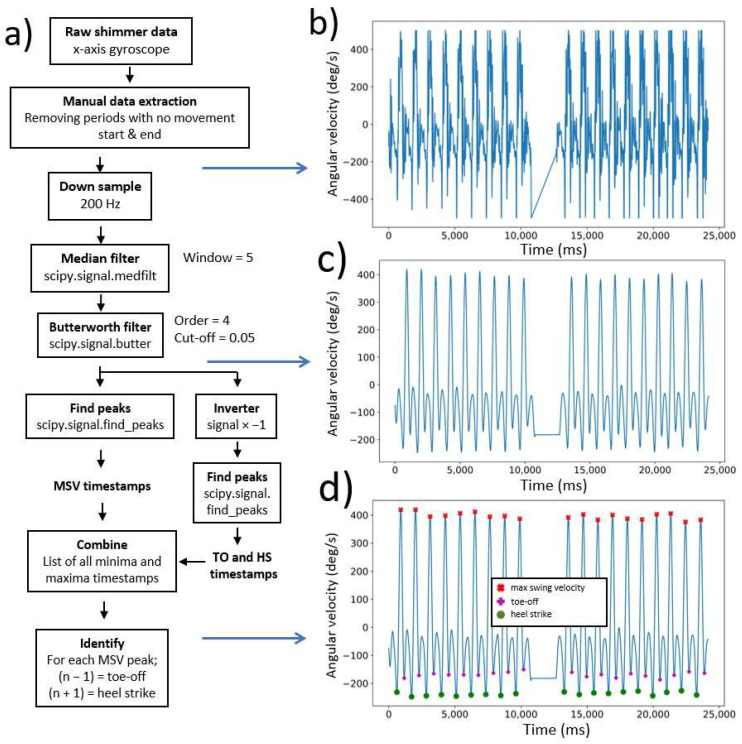
Extracting heel strike, toe-off and max swing velocity timestamps from the left ankle during a dual-task walk; (**a**) Flowchart from the data processing [raw accelerometer data to heel strike (HS), toe-off (TO) and max swing velocity (MSV)]; (**b**) Raw accelerometer data; (**c**) Filtered accelerometer data; (**d**) MSV, TO and HS peaks/troughs plotted on the filtered signal.

**Figure 2 sensors-22-05710-f002:**
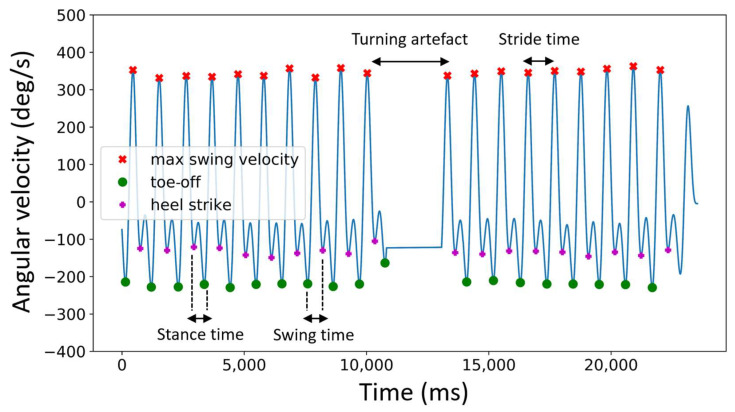
Determination of swing, stance and stride time during normal walk from the left foot.

**Figure 3 sensors-22-05710-f003:**
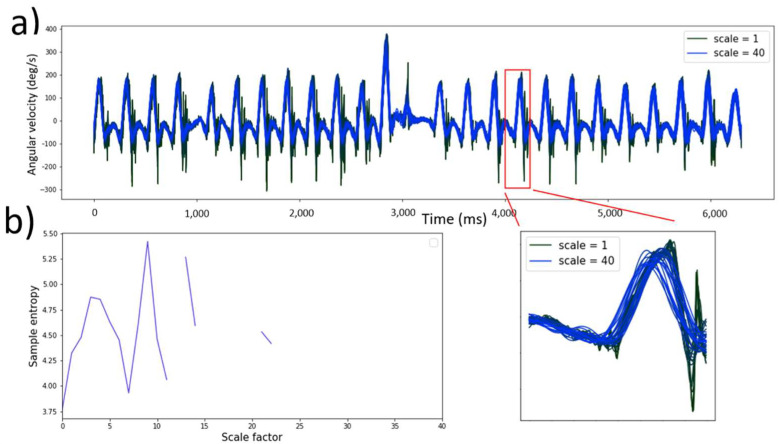
Multiscale entropy calculation of a patient during dual-task walk conditions (sample length = 2, tolerance = 0.2 and time series length = 40); (**a**) Granulating accelerometer time series where scale 1 represents the original signal and scale 40 the maximum granulated time series; (**b**) MSE curve where the area under the curve represents the complexity index.

**Figure 4 sensors-22-05710-f004:**
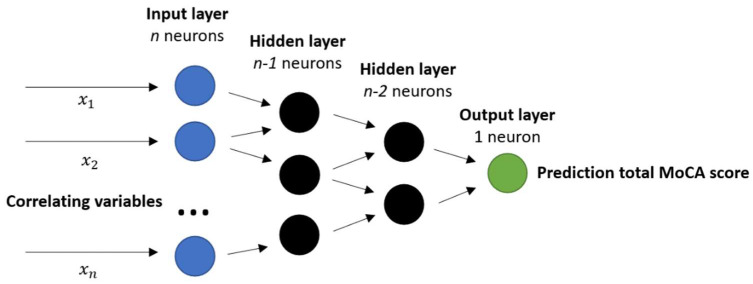
Proposed neural network for MoCA prediction with gait variables. Input layer consist of *n* discriminative gait and demographic variables for MoCA (−0.3 ≥ R² ≥ 0.3); followed up by fully connected layers with a descending amount of nodes/layer (by 1 each layer); loss function = mean squared error; optimizer = Adam; epochs = 10,000; early stopping with patience = 50.

**Figure 5 sensors-22-05710-f005:**
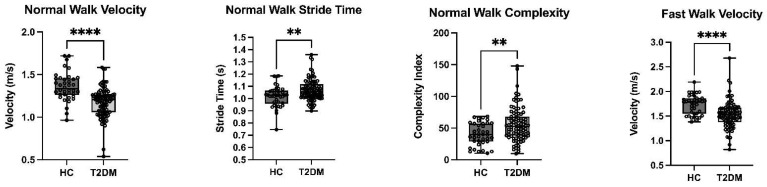
Type 2 Diabetes in Midlife is Associated with Significantly Slower Gait Speed During Normal and Fast Walking, in addition to Greater Stride Time and Greater Complexity during Normal Walking; T2DM: Type 2 Diabetes Mellitus; HC: Healthy Controls; ** *p* < 0.01; **** *p* < 0.001.

**Figure 6 sensors-22-05710-f006:**
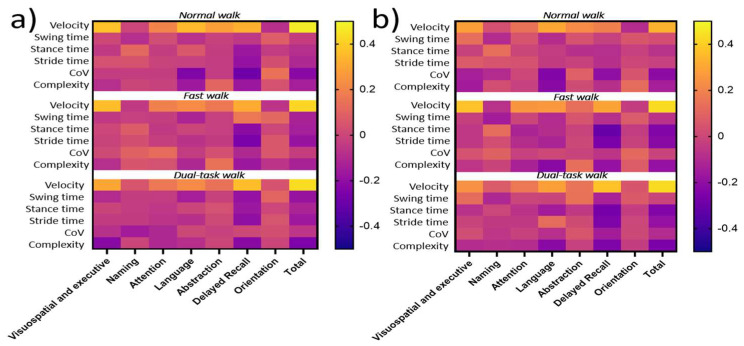
Heat map from Pearson coefficients (R²) between gait variables during normal, fast and dual-task walking conditions and MoCA scores; CoV = coefficient of variability, Complexity = Complexity index. The correlation value can vary between −1 and +1 with 0 implying no correlation; (**a**) Pearson coefficient matrix left foot and (**b**) Pearson coefficient matrix right foot.

**Figure 7 sensors-22-05710-f007:**
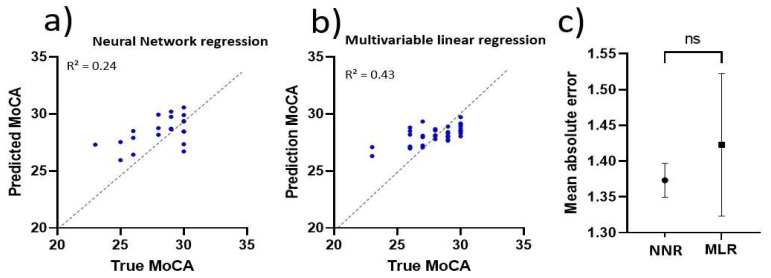
Validation of regression models; (**a**) Actual vs. prediction MoCA, predicted with neural network regression model and (**b**) Actual vs. prediction MoCA, predicted with multivariable regression model and (**c**) Comparison of mean absolute errors with 95% CI confidence intervals; NNR = neural network regression and MLR = multivariable linear regression; ns = not significant.

**Table 1 sensors-22-05710-t001:** Baseline characteristics (mean ± standard deviation) of ENBIND participants; means ± standard deviation; presented by study group. BMI: Body Mass Index.

Group	Healthy Control (*n* = 44)	T2DM (*n* = 94)	Statistic
Age	51.9 ± 8.1	52.8 ± 8.3	z = 0.47, *p* = 0.63
Sex (Female)	43% (19/44)	53% female (50/94)	z = −1.04, *p* = 0.34
BMI	26.6 ± 3.2	32.4 ± 7.8	z = 3.10, *p* < 0.05
Years of education	17.7 ± 2.2	17.2 ± 3.0	z = −1.59, *p* = 0.11

**Table 2 sensors-22-05710-t002:** Midlife Type 2 Diabetes Mellitus (T2DM) was associated with significant poorer global cognitive performance (MoCA: Montreal Cognitive Assessment).

Variable	Healthy Control (*n* = 43)	T2DM (*n* = 94)	Statistic
Visuospatial /Executive	4.9 ± 0.29	4.5 ± 0.74	z = 1.82, *p* < 0.01
Naming	3.0 ± 0	3.0 ± 0.10	z = −0.097, *p* = 0.92
Attention	6.0 ± 0	5.7 ± 0.61	z = −1.28, *p* = 0.20
Language	3.0 ± 0.21	2.8 ± 0.49	z = −1.21, *p* = 0.22
Abstraction	2.0 ± 0	1.9 ± 0.30	z = −0.99, *p* = 0.32
Delayed recall	4.2 ± 0.79	3.9 ± 1.21	z = −2.42, *p* < 0.05
Orientation	5.9 ± 0.25	5.9 ± 0.24	z = −0.26, *p* = 0.79
Total MoCA Score	29.0 ± 0.91	27.7 ± 2.1	z = −3.69, *p* < 0.001

**Table 3 sensors-22-05710-t003:** CANTAB scores for healthy controls and midlife Type 2 diabetes mellitus (T2DM); means ± standard deviation; presented by study group with appropriate analysis.

Variable	Healthy Controls (*n* = 44)	T2DM (*n* = 92)	Statistic
Paired Associates Learning—First Attempt Memory Score	12.2 ± 4.1	10.4 ± 4.5	**z = 2.21, *p* < 0.05**
Spatial Working Memory Strategy Score	8.5 ± 2.9	8.6 ± 2.5	z = −0.19, *p* = 0.84
Pattern Recognition Memory—Percentage Correct Delayed	82.4 ± 14.5	77.5 ± 14.4	z = −0.90, *p* = 0.37
Median Duration Reaction Time	409 ± 43	423 ± 54	z = 0.93, *p* = 0.35
One Touch Stockings of Cambridge—Problems Solved on First Choice	9.6 ± 3.1	8.7 ± 3.4	z = −1.47, *p* = 0.14
Rapid Visual Processing	0.89 ± 0.05	0.88 ± 0.05	z = −0.46, *p* = 0.63

**Table 4 sensors-22-05710-t004:** Gait parameters for Healthy Controls (HC) and midlife Type 2 diabetes mellitus (T2DM); means ± standard deviation; presented by study group; univariate analysis was conducted using T-tests; multivariate analysis adjusted for sex, age, BMI and years of education with coefficients for T2DM reported alongside 95% confidence intervals.

Gait Variable	HC (*n* = 44)	T2DM (*n* = 94)	t	*p*	Adj. *β* Coeff. (95% CI) for T2DM	*p*
Left Foot						
Normal Walk						
Velocity (m/s)	1.35 ± 0.16	1.12 ± 1.17	5.56	<0.001	−0.16 (−0.23, −0.09)	<0.001
Swing time (s)	0.50 ± 0.04	0.52 ± 0.05	−1.74	<0.05	0.02 (−0.00, 0.04)	0.10
Stance time (s)	0.51 ± 0.07	0.54 ± 0.07	−2.35	<0.05	0.03 (−0.00, 0.06)	0.06
Stride time (s)	1.02 ± 0.09	1.07 ± 0.09	−2.96	<0.05	0.05 (0.01, 0.08)	<0.05
Stride time variability (CoV)	3.14 ± 1.76	3.11 ± 1.51	−0.74	0.87	1.34 (−1.61, 4.30)	0.37
Complexity index	41.1 ± 17.2	55.2 ± 26.2	−3.35	<0.05	16.7 (7.23, 26.20)	<0.001
**Fast** **Walk**						
Velocity (m/s)	1.72 ± 0.19	1.53 ± 0.27	4.31	<0.001	−0.16 (−0.25, −0.06)	<0.001
Swing time (s)	0.48 ± 0.05	0.50 ± 0.07	−1.34	0.91	0.00 (−0.02, 0.03)	0.77
Stance time (s)	0.42 ± 0.07	0.44 ± 0.07	−1.75	<0.05	0.03 (−0.01, 0.06)	0.10
Stride time (s)	0.89 ± 0.09	0.94 ± 0.10	−2.43	<0.05	0.03 (−0.00, 0.07)	0.07
Stride time variability (CoV)	4.40 ± 5.14	4.32 ± 4.13	0.09	0.47	0.31 (−1.17, 1.80)	0.67
Complexity index	25.0 ± 14.1	30.5 ± 20.0	−1.86	<0.05	5.95 (−1.80, 13.70)	0.13
**Dual-Task Walk**						
Velocity (m/s)	1.34 ± 0.24	1.21 ± 0.31	2.56	<0.01	−0.08 (−0.20, 0.03)	0.16
Swing time (s)	0.54 ± 0.07	0.55 ± 0.09	−0.89	0.82	−0.00 (−0.03, 0.03)	0.91
Stance time (s)	0.53 ± 0.12	0.56 ± 0.10	−1.88	<0.05	0.04 (−0.01, 0.08)	0.13
Stride time (s)	1.07 ± 0.17	1.12 ± 0.16	−1.64	0.07	0.03 (−0.03, 0.10)	0.33
Stride time variability (CoV)	5.21 ± 3.32	5.80 ± 4.88	−0.53	0.70	0.52 (−2.22, 3.27)	0.71
Complexity index	42.1 ± 25.4	55.7 ± 34.1	−2.53	<0.05	11.14 (−1.73, 24.02)	0.09
**Right Foot**						
Normal Walk						
Velocity (m/s)	1.38 ± 0.15	1.23 ± 0.17	5.05	<0.001	−0.14 (−0.21, 0.08)	<0.001
Swing time (s)	0.52 ± 0.04	0.53 ± 0.05	−1.23	0.89	−0.01 (−0.01, 0.03)	0.26
Stance time (s)	0.50 ± 0.07	0.53 ± 0.06	−2.15	<0.05	0.03 (−0.00, 0.05)	0.07
Stride time (s)	1.03 ± 0.08	1.06 ± 0.08	−2.25	<0.05	0.03 (0.00, 0.07)	<0.05
Stride time variability (CoV)	2.59 ± 1.16	2.98 ± 1.27	0.92	0.06	0.01 (−1.88, 1.90)	0.99
Complexity index	40.9 ± 16.5	48.7 ± 22.9	−2.08	<0.05	11.65 (3.65, 19.65)	<0.001
**Fast Walk**						
Velocity (m/s)	1.76 ± 0.18	1.56 ± 0.25	4.83	<0.001	−0.17 (−0.25, −0.09)	<0.001
Swing time (s)	0.47 ± 0.04	0.49 ± 0.05	−1.47	0.93	0.01 (−0.01, 0.03)	0.24
Stance time (s)	0.41 ± 0.06	0.45 ± 0.08	−2.99	<0.05	0.03 (0.01, 0.06)	<0.05
Stride time (s)	0.89 ± 0.08	0.94 ± 0.10	−2.98	<0.05	0.05 (0.01, 0.09)	<0.05
Stride time variability (CoV)	4.54 ± 2.72	4.26 ± 2.41	−0.72	0.73	0.33 (−0.54, 1.20)	0.46
Complexity index	22.20 ± 10.8	27.9 ± 17.5	−2.09	<0.05	6.97 (1.02, 12.90)	<0.05
**Dual-** **Task Walk**						
Velocity (m/s)	1.36 ± 0.25	1.21 ± 0.27	3.51	<0.01	−0.15 (−0.25, −0.04)	<0.05
Swing time (s)	0.54 ± 0.07	0.54 ± 0.11	0.11	0.45	−0.01 (−0.04, 0.02)	0.47
Stance time (s)	0.53 ± 0.11	0.59 ± 0.14	−2.47	<0.01	0.05 (−0.01, 0.10)	0.09
Stride time (s)	1.07 ± 0.17	1.13 ± 0.18	−1.89	<0.05	0.04 (−0.03, 0.12)	0.25
Stride time variability (CoV)	5.33 ± 3.78	5.2 ± 3.56	−0.69	0.79	2.04 (−1.70, 5.78)	0.28
Complexity index	41.6 ± 26.8	51.9 ± 30.5	−2.08	<0.05	9.32 (−2.30, 20.94)	0.12

**Table 5 sensors-22-05710-t005:** Confidence intervals of parameter estimates from the multivariable regression model.

Parameter	$\beta_{1}$	$\beta_{2}$	$\beta_{3}$	$\beta_{4}$	$\beta_{5}$	$\beta_{6}$	$\beta_{7}$
Value	5.74	−4.39	1.03	0.61	1.01	1.21	21.2
CI (95%)	[1.79, 9.69]	[−8.09, −0.69]	[1.79, 3.83]	[−2.33, 3.59]	[−0.45, 2.46]	[−0.38, 2.79]	[18.74–23.68]

## Data Availability

Data available from authors on reasonable request.

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
