# Peer review of "Gait Characteristics and Cognitive Function in Middle-Aged Adults with and without Type 2 Diabetes Mellitus: Data from ENBIND"

_sensors, 2022, doi:10.3390/s22155710_

Round 1

Reviewer 1 Report

11 First the intended audience of sensors would expect much more attention to details about the sensor usage, development, application etc. I feel the paper has some merit, but might not be appropriate for the sensor audience. The author has not referenced even one sensor paper, and teh authors might not be aware of the types of papers published in this journal and teh target audience.

    2.

     Association of Type 2 Diabetes Mellitus (T2DM) to a  greater risk of dementia is not new. Also change of gait characteristics are also not new. There are many research in this area for nearly a decade that has not been included in the references. For example,

de Mettelinge, T.R., Delbaere, K., Calders, P., Gysel, T., Van Den Noortgate, N. and Cambier, D., 2013. The impact of peripheral neuropathy and cognitive decrements on gait in older adults with type 2 diabetes mellitus. Archives of physical medicine and rehabilitation94(6), pp.1074-1079.

     3. Evaluation of relationship between Gait Characteristics and Cognitive Function has been done.

     4. There are many papers even books on gait analysis with much more details rather than only angular velocity.

     5. If we want to use deep learning, as far as gait is concerned, more sensors and gait parameters would be better to capture data and may be develop new knowledge.

    6. Use of limited gait parameters as mentioned in the limitations, should have been considered in the study rather than mentioning as limitations given that research on gait analysis is quite mature.

Author Response

Response to Reviewer 1

Dear Reviewer 1 & Editorial Team at Sensors,

Many thanks for your careful, detailed review of our manuscript and for the insightful comments provided below. Based on these comments, we have made several changes to the manuscript which we believe has improved its readability for the broad readership that Sensors attracts. Please find below a point-by-point response in italic font indicating changes which have been made as a result of these observations.

  1. First the intended audience of sensors would expect much more attention to details about the sensor usage, development, application etc. I feel the paper has some merit, but might not be appropriate for the sensor audience. The author has not referenced even one sensor paper, and teh authors might not be aware of the types of papers published in this journal and teh target audience.

Many thanks for the observation. We have now added more detail about the use and development of the Shimmer™ Inertial Unit Motion (IMU) Devices, both briefly in the introduction and in more detail in the methods. We have referenced papers based on applicability to the current study and not based on journal and have submitted to sensors as this is a biomedical application of IMU technology (similar papers have appeared in Sensors before e.g PMID: 35336475, 34696131, 34696041, 34202786) to answer an important clinical question.

  1. Association of Type 2 Diabetes Mellitus (T2DM) to a  greater risk of dementia is not new. Also change of gait characteristics are also not new. There are many research in this area for nearly a decade that has not been included in the references. For example, de Mettelinge, T.R., Delbaere, K., Calders, P., Gysel, T., Van Den Noortgate, N. and Cambier, D., 2013. The impact of peripheral neuropathy and cognitive decrements on gait in older adults with type 2 diabetes mellitus. Archives of physical medicine and rehabilitation94(6), pp.1074-1079.

We have now re-emphasised our rationale to further clarify and cited the above paper. We have added a paragraph detailing previous research examining the T2DM-gait and gait-cognition link in those with T2DM in our introduction. This was not clear enough in our initial submission – thank you for your observation. 

We agree that the association between T2DM and dementia in addition to the risk between gait characteristics and dementia is not new. This is why we have discussed these links in detail in our introduction as these provide the rationale for us to examine the association between gait and cognitive function in a midlife T2DM.

Whilst you are correct to point out that some papers have examined the link between gait and cognitive performance, these are typically carried out in older adults and those with T2DM related complications such as peripheral neuropathy. We specifically targeted a midlife T2DM population (not older adults) with no T2DM complications to examine the relationship between gait performance and detailed cognitive assessment in midlife T2DM. Our aim was to detect whether this relationship is seen at this early stage, when T2DM is actually acting as a risk factor for dementia.

  1. Evaluation of relationship between Gait Characteristics and Cognitive Function has been done.

Many thanks for this observation. We agree, this is why we are assessing the relationship in midlife T2DM (in individuals free from any other T2DM-related complications). We have re-emphasised this as per point 2 above. Papers to date have focussed on later life and those with T2DM-related complications. This is in a midlife population with identification of those at greatest risk of later cognitive decline the priority.

  1. There are many papers even books on gait analysis with much more details rather than only angular velocity.

Many thanks, we have now emphasised this as a limitation in our discussion section. We were limited to those measurements available using Shimmer™ technology and have, as requested in point 1 above, added more details on Shimmer™ IMUs which are validated to compute velocity, swing, stance and stride times. They allow fewer parameters than traditional assessments such as GAITRite, but are clinically useful, pragmatic and applicable in a variety of clinical contexts. We have re-emphasised this both in the introduction and methods. Additionally, we have reflected on this in our discussion

  1. If we want to use deep learning, as far as gait is concerned, more sensors and gait parameters would be better to capture data and may be develop new knowledge.

We agree and have included this as a limitation. In the current study, we examined whether a neural network could significantly improve on traditional multivariate linear regression. Our hypothesis was to test this approach in only those gait parameters that were strongly correlated with cognition in the first instance (which was overall gait velocity across the three tasks and overall cognitive performance on the MoCA), to avoid multiple testing using these modelling approaches. Thus, we focused on those parameters that were significantly associated with cognitive performance as stated a priori.

We additionally acknowledge that the term “deep learning” may be a stretch given our study, and have instead reworded to reflect the neural networks that were used in the study.

  1. Use of limited gait parameters as mentioned in the limitations, should have been considered in the study rather than mentioning as limitations given that research on gait analysis is quite mature.

Many thanks. We agree that gait analysis in the literature is at a significantly mature stage. However, the translation of this into real-life clinical settings is not yet at such as mature stage. Our aim was in a pragmatic clinical study, to examined the utility of IMU-based gait characteristics in the assessment of the gait-cognition relationship in midlife T2DM. We have now re-emphasised this throughout our updated paper.

Reviewer 2 Report

The paper analyses the relationship between the gait characteristics derived from Inertial Motion Units (IMUs) and cognitive function assessed via the assessment tests.

Comments:

1.       Explicitly state the contribution of this study to the research field.

2.       Improve the discussion on related works on the use of the deep learning methods for gait characteristic recognition in people with cognitive decline. See, for example, „Integrated equipment for Parkinson’s disease early detection using graph convolution network“, Towards real-time prediction of freezing of gait in patients with parkinson's disease: A novel deep one-class classifier“, „Local pattern transformation based feature extraction for recognition of parkinson’s disease based on gait signals“, among others.

3.       The neural network used in this study is not deep: it has only two hidden layers and can be considered as shallow. I suggest to revise the use of the “deep learning” concept with regards to your methodology.

4.       The healthy control group had a majority of men, while the T2DM group had a majority of female. You need too asses the influence of the gender factor and distribution on the results.

5.       In some cases, the visualization of results using boxplots would present the better understanding of the data (for example, to supplement the data presented in Table 4).

6.       For Pearson correlation values also present their corresponding p-values.

7.       MoCA assessment results in ordinal scores. Since your response is ordinal then you should use ordinal regression.

8.       Check the section numbering.

Author Response

Response to Reviewer 2

Dear Reviewer 2 & Editorial Team at Sensors,

Many thanks for your careful, detailed review of our manuscript and for the insightful comments provided below. Based on these comments, we have made several changes to the manuscript which we believe has improved its readability for the broad readership that Sensors attracts. Please find below a point-by-point response in italic font indicating changes which have been made as a result of these observations.

Comments:

Explicitly state the contribution of this study to the research field.

Many thanks, this was not clear and we have now modified our discussion to reflect. Our study presents novel evidence on the association between T2DM and gait/cognitive function in midlife T2DM (the age when T2DM is acting as a risk factor for later cognitive decline). It demonstrated a significant association in middle-aged individuals with and without T2DM between cognitive and gait decrements and examined the utility of a deep-learning model to outperform standard linear regression in this regard. We have now changed our discussion to open with an explicit statement of the contribution of this work to the field.

Improve the discussion on related works on the use of the deep learning methods for gait characteristic recognition in people with cognitive decline. See, for example, „Integrated equipment for Parkinson’s disease early detection using graph convolution network“, Towards real-time prediction of freezing of gait in patients with parkinson's disease: A novel deep one-class classifier“, „Local pattern transformation based feature extraction for recognition of parkinson’s disease based on gait signals“, among others.

Many thanks. Our study concerned the application of deep learning to already-identified gait characteristics in individuals with midlife T2DM and we have re-emphasised this in the discussion. We have additionally added more detail on our use of a neural network approach, as opposed to deep learning and have amended this in our manuscript throughout.

The neural network used in this study is not deep: it has only two hidden layers and can be considered as shallow. I suggest to revise the use of the “deep learning” concept with regards to your methodology.

Thanks for this observation. We have now changed the use of “deep learning” to use of a neural network.

The healthy control group had a majority of men, while the T2DM group had a majority of female. You need too asses the influence of the gender factor and distribution on the results.

Gender has now been included in a multivariate model to assess the association of T2DM with gait performance (See Table 4) and was included in the multivariate linear regression models used to predict cognitive performance from gait velocity.

  1. In some cases, the visualization of results using boxplots would present the better understanding of the data (for example, to supplement the data presented in Table 4).

Thanks, now completed (See Figure 6, which presents those results significant after multivariate adjustment in boxplot format)

  1. For Pearson correlation values also present their corresponding p-values.

Thanks, now included in the supplementary tables.

  1. MoCA assessment results in ordinal scores. Since your response is ordinal then you should use ordinal regression.

      MoCA scores range from 0-30 and go up in one-point increments, with each increment (point) being equal. This is linear data and so linear regression was used as per previous reports analysing MoCA results.

  1. Check the section numbering

Many thanks – checked and now corrected

Round 2

Reviewer 1 Report

The authors have addressed my concern. Although I believe the addition to research is limited, but the proposed method have practical application. 

Reviewer 2 Report

The revision has been executed correctly. The quality of the manuscript has increased. I congratulate the authors for their work and recommend the article to be accepted for publication.